# Iron Speciation in Insoluble Dust from High-Latitude Snow: An X-ray Absorption Spectroscopy Study

**Shiwei Liu [1,2], Cunde Xiao [3], Zhiheng Du [1,*], Augusto Marcelli [4,5,*], Giannantonio Cibin [6], Giovanni Baccolo [7,8], Yingcai Zhu [9], Alessandro Puri [10], Valter Maggi [7,8] and Wei Xu [9]**

1   State Key Laboratory of Cryospheric Science, Northwest Institute of Eco-Environment and Resources, Chinese Academy of Sciences, Lanzhou 730000, China; liushiwei1990@lzb.ac.cn
2   University of Chinese Academy of Science, Beijing 100049, China
3   State Key Laboratory of Land Surface Processes and Resource Ecology, Beijing Normal University, 19 Xinjiekouwai Street, Beijing, 100875, China; cdxiao@bnu.edu.cn
4   Istituto Nazionale di Fisica Nucleare, Laboratori Nazionali di Frascati, 00044 Frascati, Italy
5   RICMASS, Rome International Center for Materials Science Superstripes, 00185 Rome, Italy
6   Diamond Light Source, Harwell Science and Innovation Campus, Didcot OX11 0DE, UK; giannantonio.cibin@diamond.ac.uk
7   Earth and Environmental Sciences Department, University Milano-Bicocca, 20126 Milano, Italy; giovanni.baccolo@mib.infn.it (G.B.); valter.maggi@unimib.it (V.M.)
8   INFN-Milan Bicocca Section, 20126 Milan, Italy
9   Beijing Synchrotron Radiation Facility, Institute of High Energy Physics, Chinese Academy of Sciences, Beijing,100049, China; yingcaizhu@ihep.ac.cn (Y.Z.); xuw@mail.ihep.ac.cn (W.X.)
10  CNR-IOM-OGG, c/o ESRF, 38043 Grenoble, France; alessandro.puri@esrf.fr
*   Correspondence: duzhiheng10@163.com (Z.D.); marcelli@lnf.infn.it (A.M.)

**Abstract:** Iron is thought to limit the biomass of phytoplankton populations in extensive regions of the ocean, which are referred to as high-nutrient low-chlorophyll (HNLC) regions. Iron speciation in soils is still poorly understood. We have investigated inorganic and organic standard substances, diluted mixtures of common Fe minerals in insoluble dust in snow from the Laohugou No.12 glacier, and sand (including soil and moraine) samples that were collected from western China. The speciation of iron (Fe) in insoluble dust and sand was determined by X-ray absorption near-edge structure (XANES) spectroscopy. A linear fit combination (LCF) analysis of the experimental spectra compared to a large set of reference compounds showed that all spectra can be fitted by only four species: $Fe_2O_3$, $Fe_3O_4$, biotite, and ferrous oxalate dihydrate (FOD). A significant altitude effect was detected for snow. The proportion of $Fe_2O_3$ in snow decreases gradually, and vice versa for FOD. As for $Fe_3O_4$ and biotite, the altitude effect was also detected, but separate regions should be considered to be deduced by topography. The Fe species in moraines and soils were also analyzed to identify the source of moraines and the heterogeneity of soils, and were compared with snow.

**Keywords:** Laohugou glacier; snow; insoluble dust; iron speciation; XANES and LCF

---

## 1. Introduction

Iron (Fe) contributes 5.1 mass percent to the earth's crust, and is a major component of many soil-forming parent materials. Consequently, primary or secondary Fe-containing minerals, such as olivine, biotite, pyrite, ferrihydrite, goethite, haematite, lepidocrocite, and Fe-bearing clay minerals, are significant components in most soils [1]. The presence or absence of pedogenic Fe minerals in soils, soil horizons, or soil aggregates, as well as their spatial distribution within a soil profile or an aggregate, is strongly related to the ambient physicochemical conditions, such as pH, redox potential, and activity

of organic ligands [2,3]. At variance, soil physicochemical conditions are greatly influenced by Fe-containing minerals, because they can undergo redox, sorption, (de)protonation, or (co)precipitation reactions with organic and inorganic constituents of the soil solution and the soil solid phase [4]. Because most Fe minerals are strongly colored and their presence or absence is clearly visible in soil profiles [5,6], they have been widely used to identify pedogenic processes (podzolisation, gleysation, lessivage) and to classify soils [2,4].

As a consequence of the considerable importance of Fe-containing minerals in the earth sciences, their identification and quantification in geologic materials and soils has been a major target of research, and numerous methods for the assessment of Fe minerals in soils have been developed. These methods include X-ray diffraction [7], wet chemical fractionation procedures, such as selective extraction [8–10], differential thermal analysis [11], Mössbauer spectroscopy [7], and colorimetry [5,6]. However, most of these methods are either applicable only to crystalline phases (XRD) or provide only operationally defined results (dissolution methods). Moreover, none of these methods provides information on the micro-morphological arrangement of different Fe species in natural soil structures, e.g., aggregates.

Aerosols originating from soils of arid regions are probably the main source of bioavailable Fe in the open ocean. However, recently, a more complex scenario, with multiple environmental sources for aerosol containing iron, is emerging [12]. Although aerosol solubility could be heavily affected by Fe speciation, which in turn could vary considerably among possible aerosol source environments, no such relationship has yet been clearly established, largely owing to a lack of direct measurements of particle speciation. Synchrotron radiation (SR) is an ideal X-ray source that provides a high intensity photon for the investigation of Fe speciation.

X-ray fluorescence (XRF) and X-ray absorption near-edge structure (XANES) spectroscopy are major SR techniques. XANES spectroscopy provides information on the valence state and binding structure of absorbing elements. XANES was first shown to be determined by multiple scattering resonances of the photoelectron that is emitted at the absorbing atomic site by Belli et al. [13], who probed the local structure of a nanoscale cluster of atoms centered at the absorbing atomic site. Bianconi et al. [14] used XANES spectroscopy to quantitatively probe into subtle local lattice distortions of the Fe site. XANES spectroscopy is widely used as a tool for local structure investigations [15], and has been used for chemical state analyses on a wide range of Fe elements [16–19].

A linear combination fit (LCF) analysis has previously been chosen to directly determine the substance speciation in soil, aerosol, and sediment [19–22]. This technique has already been applied to the investigation of the local molecular structure in the region of selected metal atoms in homogenous dust molecules. In the present study, XANES and LCF were applied to investigate the Fe speciation in dust dispersed in snow on the LHG glacier, which is located in western China. Several local and foreign soils were also measured in an attempt to identify the dust's source.

## 2. Materials and Methods

### 2.1. Sample Collection

The Laohugou (LHG) NO.12 glacier (5Y448D0012) is one of the largest valley glacier groups on Qilian Mountain, which is located at the northern edge of the Tibetan Plateau adjacent to the Gobi desert. The LHG No.12 glacier is a typical continental glacier [23] that is located on the north slope of western Qilian Mountain. It covers an area of 21.9 km$^2$ with a large accumulation zone. Its terminal elevation is 4260 m above sea level (a.s.l.). Above 4500 m a.s.l., the glacier is flat, while below 4500 m a.s.l., a large serac area is present.

Fresh surface snow samples were collected at the site shown in Figure 1 with a 500 mL LDPE bottle at different altitudes (4300 m, 4400 m, 4500 m, 4600 m, 4700 m, and 4900 m) on 25 May 2016. In the laboratory, the collected snow samples were melted rapidly at room temperature (~25 °C). After melting, all samples were filtered immediately to minimize the adhesion of particles to the surface of the bag by pouring the snow-sourced water into a 500 mL percolator and pumping the

sample slowly through an oven-dried (at 60 °C for 24 h), 70 μm plastic filter. After, the filters were removed from the holders and placed into plastic captures, which held them in place under a dry and low temperature condition. The plastic filters with dust were used for the XRF experiments. Besides this, one soil sample from the end of the LHG glacier (4200 m a.s.l.) and two moraine samples from the LHG tongue at ~4250 and 4700 m a.s.l. were excavated. In order to compare the snow samples with LHG soil, and trace the source of the insoluble content in snow, several soil and sand samples around the LHG region were also collected. Table 1 summarizes the information on the samples. The moraine, soil, and sand samples were dried at room temperature (~25 °C) for a few days, then powdered and filtered by a 70 μm agate filter. We then prepared pellets of ~2 g each for the X-ray absorption measurement.

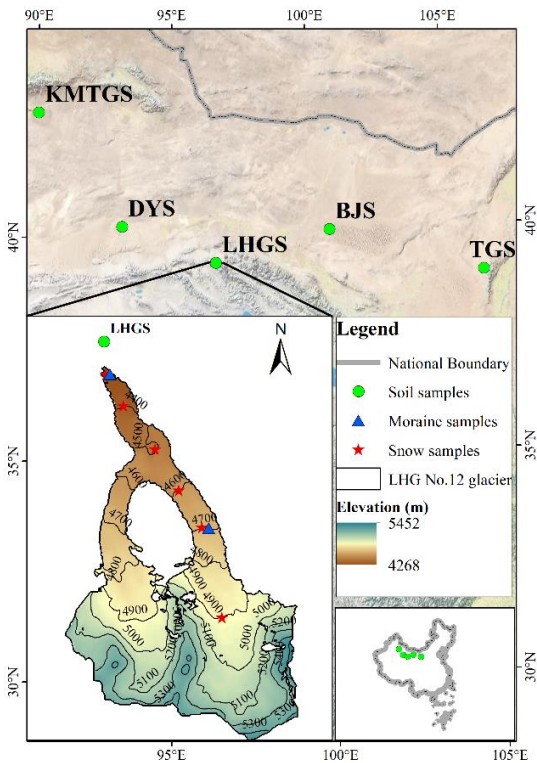

**Figure 1.** The location of the Laohugou (LHG) glacier and the collection sites of soils and sands described in Table 1.

**Table 1.** The collected samples.

| Sample Name | Substance Configuration | Collecting Location |
|---|---|---|
| LHGS | soil | End of the LHG No.12 glacier |
| KMTGS | sand | Kumtag desert |
| DYS | soil | Dunhuang yadan |
| TGS | sand | Tengger desert |
| BJS | sand | Badain Jaran desert |
| Tongue moraine | moraine | 4250 m a.s.l. LHG No.12 glacier |
| 4700 m moraine | moraine | 4700 m a.s.l. LHG No.12 glacier |
| 4300 m snow | insoluble dust in snow | 4300 m a.s.l. LHG No.12 glacier |
| 4400 m snow | insoluble dust in snow | 4400 m a.s.l. LHG No.12 glacier |
| 4500 m snow | insoluble dust in snow | 4500 m a.s.l. LHG No.12 glacier |
| 4600 m snow | insoluble dust in snow | 4600 m a.s.l. LHG No.12 glacier |
| 4700 m snow | insoluble dust in snow | 4700 m a.s.l. LHG No.12 glacier |
| 4900 m snow | insoluble dust in snow | 4900 m a.s.l. LHG No.12 glacier |

## 2.2. Choice of Reference Compounds

The X-ray Fluorescence (XRF) lines of each element are defined by the transition energy among different orbitals. Each line occurs at a fixed energy and the fluorescence lines are clearly separated for different elements. The energy value of each emission line allows us to identify the element in the investigated sample. For instance, the $K_\alpha$ and $K_\beta$ lines of iron can be detected at 6.4 keV and 7.06 keV, respectively. The $L_\alpha$, $L_\beta$, and $L_\gamma$ lines of lead occur at 10.6 keV, 12.6 keV, and 14.8 keV, respectively. Usually, one should at least rely on two lines to justify the existence of the corresponding element because, due to a poor energy resolution, the XRF lines of some elements may overlap and cannot be distinguished. For instance, the $K_\alpha$ line of arsenic is quite close to the $L_\alpha$ line of lead.

## 2.3. X-ray Absorption Measurements

The iron speciation in the samples was determined using synchrotron-based techniques, such as X-ray Absorption Near-Edge Structure (XANES) spectroscopy and microscopic X-ray fluorescence measurements at LISA (Linea Italiana per la Spettroscopia di Assorbimento di raggi X), the BM08 beamline of the European Synchrotron Radiation Facility (ESRF). The energy has been selected with a double crystal monochromator using a Si(111) crystal pair and two Si-coated mirrors for harmonics rejection. Further details about the beamline and the technical specifications are available elsewhere [24,25]. We worked at room temperature, and, due to the flux available on the bending magnet beamline, we never observed radiation damage to our samples.

The preliminary Fe K-edge and XRF data results indicate that they can be used to trace the variability of mineralogy on snow, ice, and solid samples with different provenances. Energy was calibrated by setting to 7112 eV the energy position of the first inflection point in the K-edge of a Fe foil recorded in a double transmission setup. For each snow sample and solid sample, one to four scans were recorded, depending on the sample content and Fe concentration. Spectra have been collected under vacuum (10–2 mbar) at ambient temperature in fluorescence mode. We used a four-channel SDD (Solid State Detector) detector with an energy resolution of ~180 eV at the Fe k$\alpha$ emission line (~6400 eV). In the case of filters, due to the large dimensions and the inhomogeneity, an unfocused beam of ~2 mm$^2$ was used to probe these samples. Data were averaged if necessary and normalized using the Athena [26] software. EXAFS spectra were then background subtracted from these normalized data using the XAFS code [27].

## 3. Results and Discussions

### 3.1. Fe K-Edge XANES of Standards

Ten iron standard natural compounds, one mineral, and one organic iron system were selected as references. Figure 2 shows the XANES spectra at the iron K-edge of the 12 standard substances listed in Table 1. The standard substances in which iron is in the Fe (II) state include $FeSO_4$, $FeCl_2$, $FeS_2$, and FeS, and those in which iron is in the Fe (III) state are $FeCl_3$, $FePO_4$, $Fe_2O_3$, $Fe(SO_4)_3$, and $Fe(NO_4)_3$, including the $Fe_3O_4$. These standards showed clear differences in the energy positions of the centroids of their normalized pre-edge peaks, of the energy position, and of their white lines (Table 1). Typically, the Fe absorption edge shifts upward from 7120.5 to 7128.6, increasing the oxidation state. The edge position of Fe (II) and Fe (III) compounds falls in the ranges 7116–7121 eV and 7126–7129 eV, respectively. According to Figure 2b, an intense pre-edge peak is detected in $FeS_2$, FeS, $Fe_3O_4$, and $Fe_2O_3$. The pre-edge position for $Fe_3O_4$ occurs at a lower energy than that for $Fe_2O_3$, biotite, and ferrous oxalate dihydrate (FOD).

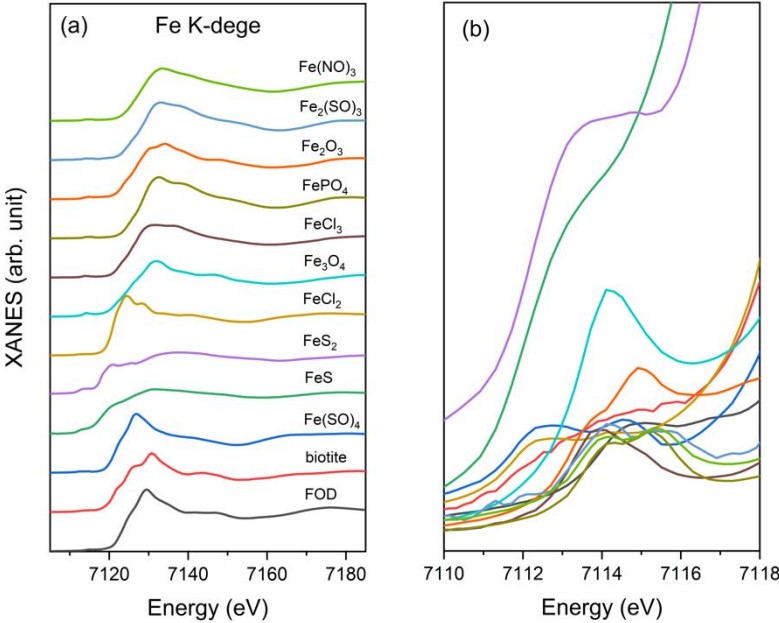

**Figure 2.** (**a**) A comparison of the XANES spectra of iron standards (**b**) and the pre-edge region of the iron K-edge of the spectra shown in the left panel.

### 3.2. Fe K-Edge XANES of the Soil, Moraine, and Snow Samples

Fe speciation is of great interest due to its role in the dissolution process of Fe from dust minerals and for the possibility to probe into the contribution of soluble iron to the ocean. Indeed, Fe is correlated to phytoplankton growth in the sea and affects the oceanic biogeochemical cycle [28,29]. It is, thus, fundamental to probing and understanding the different factors that affect Fe speciation at a regional scale, to further constrain the role of iron in biogeochemical models [30].

The Fe-edge XANES spectra of the soil of the LHG glacier (LHGS sample) and the moraine samples are shown in Figure 3a, and those of the other relevant soil samples and of the six snow samples are shown in Figure 3b,c, respectively. The result indicates that biotite and $Fe_2O_3$ are the main components in these samples. Although the Fe speciation in biotite is $Fe^{2+}$, the Fe oxide ($Fe_2O_3$) in these samples indicates that $Fe^{3+}$ is the primary component because $Fe^{3+}$ is much more stable. For comparison, the spectra of two analogous reference compounds, i.e., Fe(II)-biotite and Fe(III)-$Fe_2O_3$, have been included in all figures. Moraine is the sediment that is transported by the glacier accumulation process, which mainly comes from a fragment of a mountain. The Fe-edge XANES spectra of the moraines and LHGS are similar and closely resemble biotite, indicating that the iron in both the LHGS and moraine is primarily associated with biotite. This provides another explanation for the moraine source on the glacier, although at the molecule level. Figure 3b compares the Fe-edge XANES spectra of relevant soils and sands surrounding the LHG glacier. The spectra of soil and sand samples resemble those of biotite and $Fe_2O_3$, but are a complex mixture of iron minerals, and not just a mixture of biotite and $Fe_2O_3$.

The result makes it possible to distinguish the LHGS sample. Figure 3c shows the Fe-edge XANES spectra of the snow samples and relevant molecules of the iron element. The spectra of the snow samples are homologous but inconsistent with the spectra of biotite and $Fe_2O_3$, pointing out that the insoluble dust of the snow samples may not come only from a local dust source (i.e., the LHGS or moraine), but may contain other reference compounds.

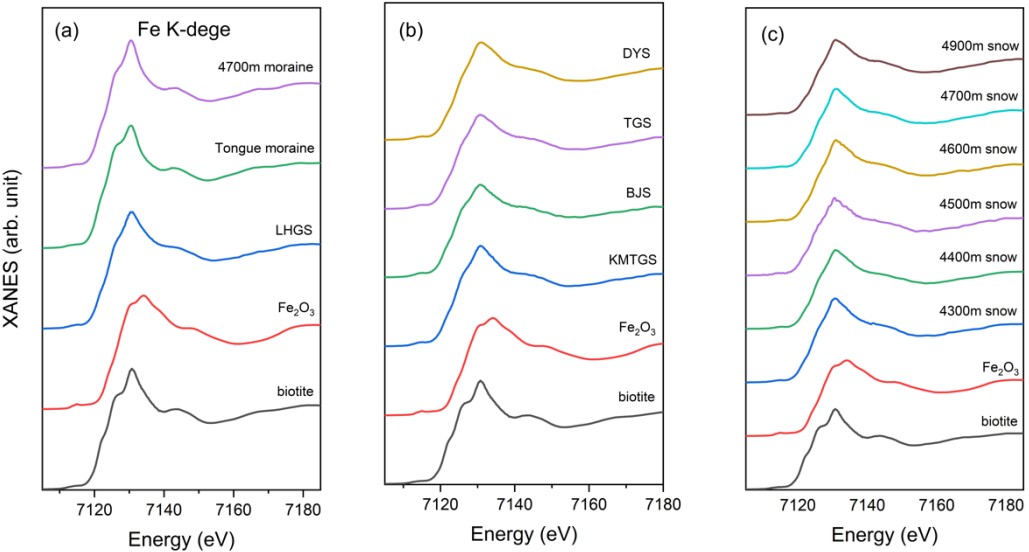

**Figure 3.** XANES spectra at the iron K-edge of (**a**) the soil and moraine samples; (**b**) the reference soil; and (**c**) the snow samples.

### 3.3. Linear Combination Fitting

The $k^2$-weighted Fe K-edge XANES spectra of all samples were analyzed by a principal component analysis (PCA) [20] to determine the number of reference model spectra needed to simulate the experimental data. Based on the PCA results, a target transformation (TT) was further performed in order to select the reference standard substances that were most likely present in these samples [31]. After that, based on the results from the PCA/TT, a linear combination fitting (LCF) analysis was performed using the software Athena to calculate the proportion of each iron reference in the soil, moraine, and snow samples. Several standard substances were selected to represent the possible iron compounds that were potentially present in the LHG samples. The PCA/TT result shows that $Fe_2O_3$, $Fe_3O_4$, biotite, and ferrous oxalate dihydrate (FOD) are the major standard substances in the soil samples, moraine samples, and snow samples. The four reference compounds were used to run the LCF analysis. A fit range of −20 to 50 eV was selected to fit the sample spectra. The LCF results are summarized in Table 2.

**Table 2.** The linear combination fitting (LCF) results of the soil, moraine, and snow samples. FOD, is the ferrous oxalate dihydrate.

| Samples | $Fe_2O_3$ | $Fe_3O_4$ | Biotite | FOD | R-Factor |
|---|---|---|---|---|---|
| | Proportion (Uncertainty) | | | | |
| LHGS | 0.080 (0.025) | 0.218 (0.019) | 0.524 (0.013) | 0.178 (0.008) | 0.0002 |
| Tongue moraine | / | / | 0.738 (0.089) | 0.262 (0.102) | 0.0017 |
| 4700 m moraine | / | / | 0.719 (0.018) | 0.281 (0.018) | 0.0022 |
| KMTGS | 0.307 (0.061) | 0.354 (0.038) | 0.196 (0.034) | 0.143 (0.018) | 0.0010 |
| DYS | 0.346 (0.031) | 0.235 (0.026) | 0.338 (0.014) | 0.081 (0.008) | 0.0003 |
| TGS | 0.192 (0.018) | 0.197 (0.015) | 0.417 (0.010) | 0.194 (0.005) | 0.0002 |
| BJS | 0.151 (0.035) | 0.147 (0.026) | 0.514 (0.019) | 0.188 (0.009) | 0.0005 |
| 4300 m snow | 0.478 (0.094) | 0.208 (0.148) | / | 0.315 (0.050) | 0.0075 |
| 4400 m snow | 0.264 (0.058) | 0.197 (0.043) | 0.275 (0.027) | 0.264 (0.016) | 0.0006 |
| 4500 m snow | 0.248 (0.053) | 0.092 (0.042) | 0.479 (0.026) | 0.181 (0.019) | 0.0008 |
| 4600 m snow | 0.200 (0.114) | 0.479 (0.071) | / | 0.320 (0.036) | 0.0040 |
| 4700 m snow | 0.194 (0.036) | 0.187 (0.099) | 0.278 (0.028) | 0.341 (0.017) | 0.0009 |
| 4900 m snow | 0.168 (0.048) | / | 0.468 (0.027) | 0.365 (0.034) | 0.0044 |

The spectra of all soil, moraine, and snow samples closely resemble a combination of $Fe_2O_3$, $Fe_3O_4$, biotite, and FOD. The spectra of the LHG moraines are closer to biotite and FOD, and the overall curvature of the Fe-XANES spectrum for the moraines is similar to that for biotite, which demonstrates that the two reference compounds are the components of moraine. As shown in Figure 4, the LCF result shows that biotite is the major component of both moraines with a proportion of 73.8% and 71.9% in the LHG tongue and 4700 a.s.l. samples, respectively. At the same time, the minor component is FOD with a proportion of 26.2% and 28.1%, respectively. The two moraine samples come from mountain soil and debris, which accumulate by the glacier's movement.

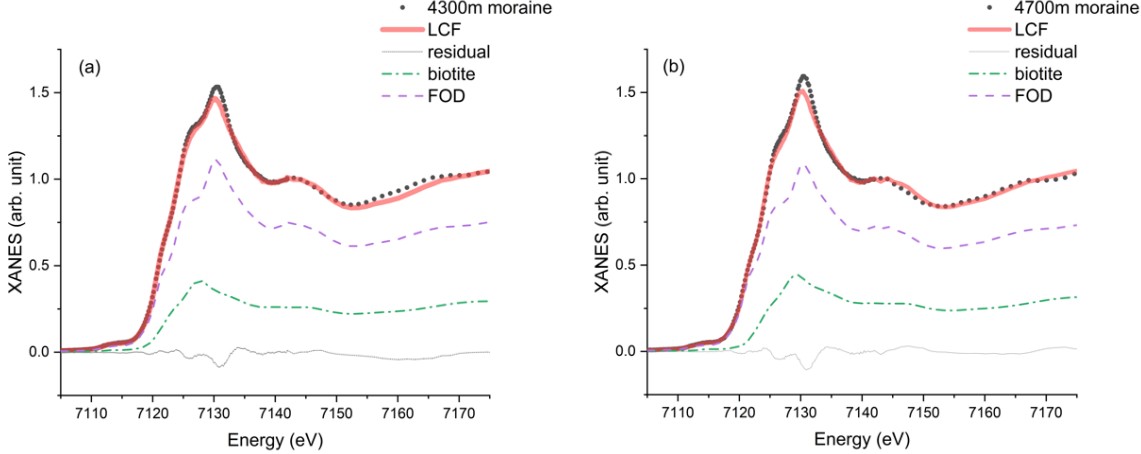

**Figure 4.** Iron speciation in the moraine samples, as calculated by the linear combination fitting (LCF) of Fe K-edge XANES spectra of two moraine samples collected at 4300 m (**a**) and at 4700 m (**b**).

There is a different fraction of reference compounds between the LHGS and LHG moraines, which means that it is difficult to be sure that all the moraines come from local soil. The LHGS sample was collected at the end of the LHG No.12 glacier, which is 4200 m a.s.l.. The LCF analysis points out that biotite is the major component, with 52.4% of reference compounds. The difference among our soil samples and moraines could be due to the heterogeneity from the mountain soils. However, spectral features and the LCF analysis point out that the LHGS sample and the moraines have a similar iron composition, which provides another explanation for the moraine source on the glacier, although at the molecule level. The Fe speciation can distinguish local soil samples (LHGS) from potential desert sources (i.e., KMTGS, DYS, TGS, and BJS) using the LCF method (Table 3). The LCF spectra of the LHGS and KMTGS samples are shown in Figure 5. The other samples were also fitted, but are not shown.

**Table 3.** The energy positions of the pre-edge peak centroid, the inflection point of the absorption edge, and the white line of the Fe K-edge XANES spectra acquired for different Fe-bearing standards.

| Compound | Fe Oxidation State | Centroid Pre-Edge Peak | Inflection Point Absorption Edge | White Line |
|---|---|---|---|---|
| | | | Energy Position/eV | |
| biotite | +2 | 7114.9 | 7121.0 | 7127.9 |
| FOD | +2 | 7115.1 | 7121.0 | 7127.9 |
| $FeSO_4$ | +2 | 7114.6 | 7120.5 | 7127.0 |
| FeS | +2 | 7113.2 | 7116.4 | 7131.8 |
| $FeS_2$ | +2 | 7114.8 | 7116.6 | 7120.8 |
| $FeCl_2$ | +2 | 7114.2 | 7119.4 | 7124.5 |
| $Fe_3O_4$ | +2/+3 | 7114.2 | 7128.3 | 7132.0 |
| $FeCl_3$ | +3 | 7114.1 | 7126.3 | 7131.9 |
| $FePO_4$ | +3 | 7115.2 | 7127.2 | 7132.6 |
| $Fe_2O_3$ | +3 | 7114.9 | 7125.9 | 7134.1 |
| $Fe(SO_4)_3$ | +3 | 7114.2 | 7128.6 | 7133.3 |
| $Fe(NO4)_3$ | +3 | 7115.2 | 7127.3 | 7133.5 |

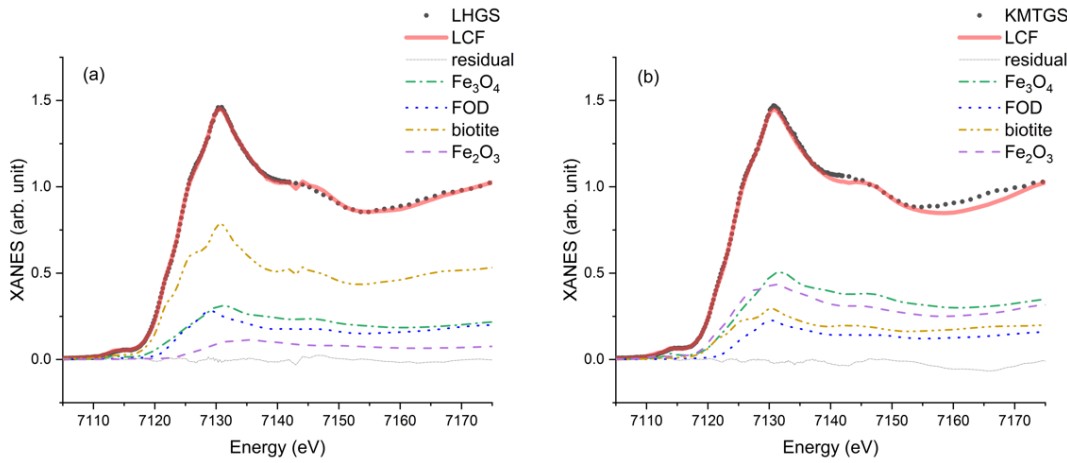

**Figure 5.** The iron contribution to the desert samples that were collected at different regions, as calculated by linear combination fitting (LCF) from the Fe K-edge XANES spectra of the LHG No.12 glacier (**a**) and the Kumtag desert (**b**).

The fraction of biotite in the LHGS and in the BJS is comparatively the same, more than 50%, which is the highest fraction, and is followed by the TGS, DYS, and KMTG with a proportion of 41.7%, 33.8%, and 19.6%, respectively. The ratio of $Fe_2O_3$ to $Fe_3O_4$ in the endemic soils is almost equal to one. However, the proportion of $Fe_2O_3$ in the LHGS is 8%, whereas the proportion of $Fe_3O_4$ is 21.8%. That makes it possible to separate the LHG from the ecdemic soils within the local source and the far source contributions of the insoluble dust to the snow precipitation. For the FOD, almost all of the soil samples are similar, except for DYS. They contain less FOD than the LHG moraines, probably due to the dry conditions and the different soil texture. The proportion of reference compounds in all measured soil (sand) samples is shown in Figure 6. A slow change in the proportion is observed and makes it possible to identify different sources of the insoluble dust in the LHG snow samples.

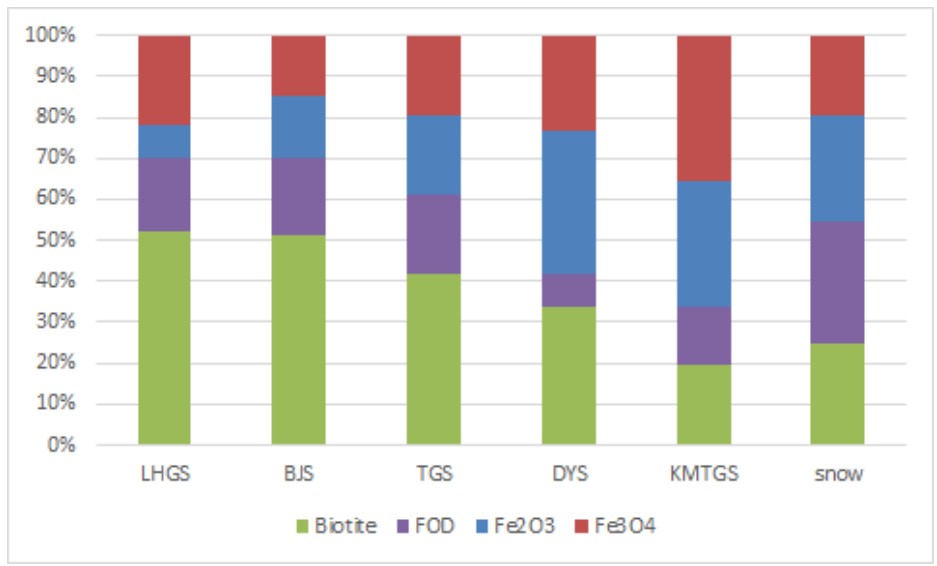

**Figure 6.** The different proportions of the four iron components in both the soil and snow samples showed in Figure 5.

Figure 7 shows the Fe K-edge XANES spectra of snow samples from different altitudes (4400 m and 4700 m a.s.l. are shown in (a) and (b), respectively) in the LHG glacier together with the Fe reference compounds that yield the best fits by the LCF analysis. The R-factor of LCF indicates that the fit is acceptable, and half of the six snow samples are composed of four iron components, except for

the 4300 m, 4600 m, and 4900 m snow samples, for which the LCF performed better with only three reference components.

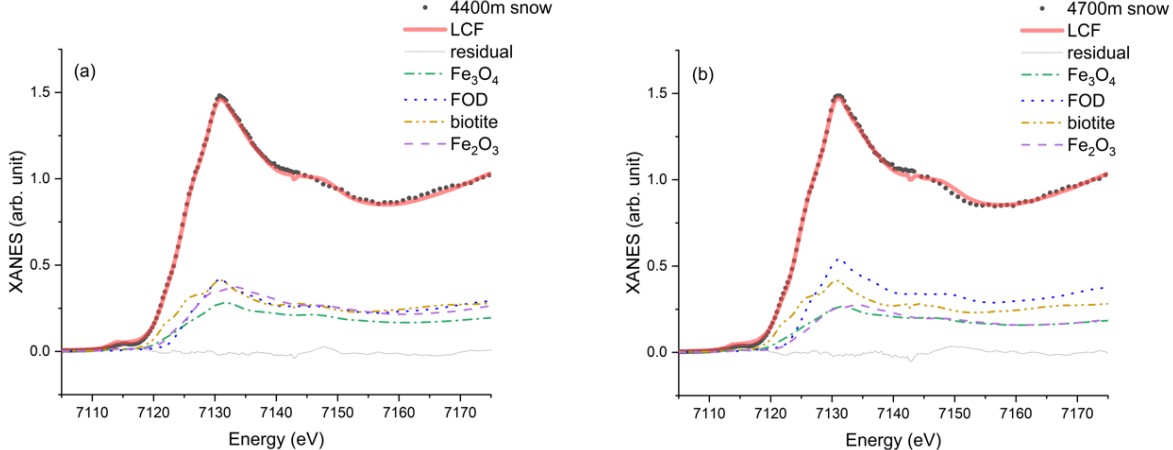

**Figure 7.** Iron speciation in different altitude snow samples, as calculated by linear combination fitting (LCF) from the Fe K-edge XANES spectra of the snow samples collected at 4400 m (**a**) and at 4700 m (**b**).

The proportion of iron components in the snow samples is shown in Figure 8. $Fe_2O_3$, $Fe_3O_4$, biotite, and FOD share a comparable proportion in the average of all snow samples, with the average percentage of 25.9%, 19.4%, 25.0%, and 29.8%, respectively. According to Figure 8, these snow samples can be separated into two sets of altitudes (i.e., 4300–4500 m and 4600–4900 m). In both categories, the snow samples contain more biotite and less $Fe_2O_3$ and $Fe_3O_4$. At a lower altitude, biotite is not a reference component in the 4300 m snow sample, and the content of $Fe_2O_3$, $Fe_3O_4$, and FOD is 47.8%, 25.8%, and 31.5%, respectively. With an increase in the elevation, biotite becomes a major component (up to 47.9%), and the proportion of the other reference components decreases.

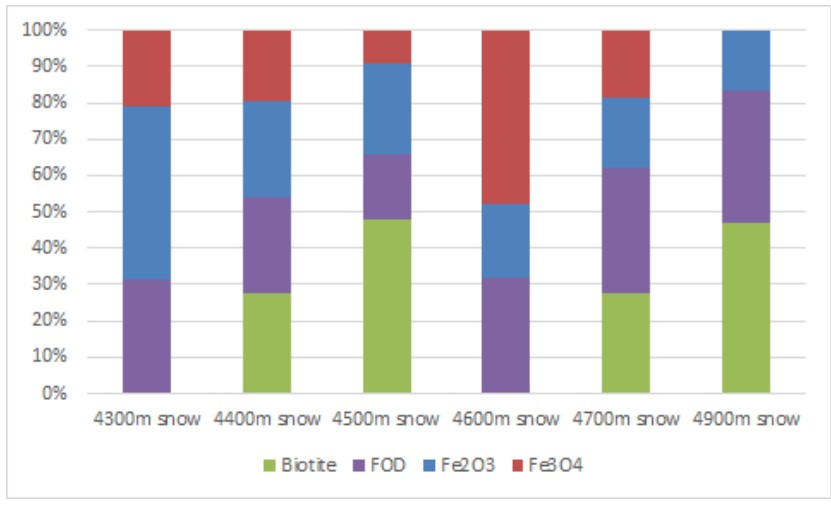

**Figure 8.** The proportion of the four iron components in the snow samples collected at different altitudes.

The average proportion of $Fe_2O_3$, $Fe_3O_4$, biotite, and FOD at a lower altitude by LCF is 33%, 16.6%, 25.1%, and 25.3%, respectively. For the higher altitude region, biotite is also not a component. In the lowest altitude snow sample, $Fe_3O_4$ is the major component. With an increase in the elevation, biotite becomes the major component (up to 36.5%), and the proportion of the other reference components is reduced. Moreover, $Fe_3O_4$ disappears at 4900 m. The average proportion of $Fe_2O_3$,

$Fe_3O_4$, biotite, and FOD in the lower altitude snow samples, as calculated by LCF, is 18.8%, 22.2%, 24.9%, and 34.2%, respectively.

When all of the regular patterns of reference components in the snow samples collected at different altitudes are combined, a clear effect versus altitude emerges. The proportion of $Fe_2O_3$ in the snow samples decreases gradually. The proportion of FOD in the snow samples clearly decreases at low altitude (4300–4550 m a.s.l.) and increases weakly at high altitude (4600–4900 m a.s.l.). As for $Fe_3O_4$ and biotite, an altitude effect could also be detected; however, separate regions should be considered by topography. The absence of biotite and $Fe_3O_4$ in several snow samples makes it difficult to confirm the source of precipitation and the mechanism that is associated with the altitude. A deeper investigation is in progress, and additional soil and snow samples will be investigated.

## 4. Conclusions

This study investigates the speciation of Fe in insoluble dust and sand samples from western China by means of X-ray absorption spectroscopy techniques and in particular by XANES spectroscopy. The insoluble dust in snow and sand samples represents the contribution of typical arid regions. The linear combination fit of experimental XANES spectra of selected relevant Fe compounds suggests that Fe in the insoluble dust is mainly present as inorganic species, e.g., $Fe_2O_3$, $Fe_3O_4$, biotite, and ferrous oxalate dihydrate. Biotite is the major component in the LHG soil and moraines, and appears also to be a significant component in the snow samples. There is an altitude effect of the reference components on the insoluble dust in snow: the proportion of $Fe_2O_3$ in the snow decreases gradually, and the proportion of FOD in the snow clearly decreases at low altitude (4300–4550 m a.s.l.) and increases weakly at high altitude (4600–4900 m a.s.l.). As for $Fe_3O_4$ and biotite, the altitude effect could also be detected, but a better analysis of the investigated region should be considered as being deduced by topography.

The spectra of the LHG moraines are closer to biotite and FOD, and the overall shape of the XANES spectra at the Fe K-edge for moraines is similar to the biotite spectrum. The results demonstrate that the two reference compounds are the major components of moraine, and that the LHGS and moraines have a similar iron composition. The data provide an alternative explanation for the moraine source in the glacier for different Fe contents and valences. A gradual change in the composition is evident in the soil samples from local and long-distance sources. However, additional work on Fe composition and speciation in other samples is necessary to better understand the mineral dust transport mechanisms.

**Author Contributions:** S.L., C.X., Z.D., A.M., and W.X. designed the study. Z.D. and S.L. performed the field work. S.L., Z.D., A.M., G.C., A.P., G.B., and Y.Z. performed the experiments at ESRF. S.L., Y.Z., Z.D., and W.X. performed the XAS data analysis. All authors wrote the manuscript and contributed to the discussion/interpretation of the results.

**Funding:** This research was funded by Cunde Xiao grant number 41425003, Zhiheng Du grant number 41701071, Wei Xu grant number U1532128, and the CAS "Light of West China" Program.

**Acknowledgments:** We strongly acknowledge the staff of the Italian CRG LISA for their support at ESRF within the experiment 08-01-1031 on Beamline BM08 and for many fruitful discussions. The support of DARA Department of the Italian *Presidenza del Consiglio dei Ministri* is gratefully acknowledged.

**Conflicts of Interest:** No conflicts of interest.

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
