# Peer review of "Iron Speciation in Insoluble Dust from High-Latitude Snow: An X-ray Absorption Spectroscopy Study"

_condensedmatter, doi:10.3390/condmat3040047_

Round 1

Reviewer 1 Report

The paper of Liu et al. demonstrates Fe speciation in environmental specimens from north-western China, including water from a glacier. The scientific case is very interesting, with a strong impact to environmental sciences, ultimately aiming to understand the dependence of Fe oxidation state on different environmental conditions. The research was well conducted and well documented, however some experimental aspects have to be stressed more precisely. Generally speaking, the paper is well written, the language is correct, with an exception of grammar mistake, page 5, line 148 that has to be fixed. The experimental part from the ESRF, section 2.2 - 2.3, and data-processing deserves to be completed with the following:
1. What was the physical form of pieces of sand, soil, moraine investigated with X-rays? Did you make any pellets? How much material did you probe? How much is the absolute and relative (ppm) abundance of Fe in the material? I understood the dust from snow was probed from the sediment deposited on a filter, isn't it?
2. How did you do the experiment? What were the experimental conditions? Did you work in ambient temperature or with a cooling system (e.g. a cryostat, cryojet) to prevent radiation damages?
3. What detectors did you use to collect XANES spectra? If any energy-dispersive detector was used, what was its energy resolution? How did you take into account matrix effects to correct the XANES spectra? How much time did you spend to record a complete XANES spectrum?
4. What was the beam size? Did you do a bulk or a local analysis? Did you illuminate the full sample with an unfocussed beam or did you probe Fe XANES spectra from hotspots with a focused beam?
Once the issues mentioned above are properly addressed, the paper is recommended for publication. At this moment I am suggesting a minor revision.

Reviewer 2 Report

The manuscript reports results of the Fe chemical speciation analysis from fresh snow samples collected at different altitudes from the surface of a glacier, moraines and soil/sand samples from arid areas surrounding over a large geographical area the glacier, all collected from western China. The manuscript aims to contribute in a better understanding of Fe long transportation and accumulation mechanisms, in particular those that might be responsible for the presence of bioavailable Fe in the open ocean. In this respect, there is no adequate connection/discussion of the present results with the final objective of the present work. It is suggested that the authors should further enrich their discussion and conclusions section of the manuscript. Moreover, a key conclusion of the manuscript, namely that there is an an altitude effect of reference Fe2O3/FOD components in snow insoluble dust with the proportion of Fe2O3 to decrease gradually, whereas FOD increases, it is not supported adequately by the results (see Table 3 and Fig. 8). I suggest that the authors should provide more convincing arguments in support of this statement. For example, considering the LCF results presented in the Table 3 and the associated uncertainties, it seems that FOD does vary too much, whereas Fe2O3 shows rather a random variation versus altitude. On the other hand, an interesting finding of the present work is that LHGS and moraines samples have a similar iron composition that it is different from that of the fresh snow samples.

The chosen analytical methodology for Fe chemical speciation is adequate, although poorly described in some aspects (see below more specific comments), however the data analysis procedure that has been followed is very robust and has provided reliable results.

Some suggestions are proposed below to improve the clarity of the experimental XANES part:

1) The basic features of the ESRF beamline and experimental set-up should be reported, monochromator, energy resolution, etc.

2) How the XANES measurements were carried out? in transmission or in fluorescence mode? It is mentioned that the Fe foil XANES spectrum was recorded in double transmission setup. Was this the case for the rest of the samples? If yes, how the solid soil/sediment/sand samples were prepared (sample mass, area, etc.)

3) How much was the estimated mass of the fresh snow samples deposited on the plastic filter?

Round 2

Reviewer 1 Report

Although it is now to me satisfactory how the authors had addressed the issues raised in my 1st review report, it's really a pity that their valuable explanation (i.e. the text in red in the rebuttal) was not incorporated into the manuscript. If a major part of rebuttal was transferred to the manuscript, its quality would be significantly improved. But now, the important aspects of experiment/ project are still missing. At this moment I will leave the paper with the recommendation of "minor revision" and I am strongly suggesting to enrich the manuscript with the your explanation that you already gave in the rebuttal.

Reviewer 2 Report

The revised manuscript can be accepted for publication

Author Response

Dear Reviewer: 

Thank you very much for the comments concerning out manuscript entitled "Iron speciation in insoluble dust from high-latitudes snow: an X-ray absorption spectroscopy study" (ID: condensedmatter - 373705).

Yours sincerely.